# Overwintering Ecology and Novel Trapping Strategies for Sustainable Management of the Common Pistachio Psyllid (*Agonoscena pistaciae*) in Pistachio Orchards

**DOI:** 10.3390/insects16111150

**Published:** 2025-11-10

**Authors:** Bülent Laz

**Affiliations:** Department of Forest Engineering, Faculty of Forestry, Kahramanmaraş Sütcü Imam University, 46050 Kahramanmaras, Türkiye; bulentlaz@ksu.edu.tr

**Keywords:** pistachio psyllid, *Agonoscena pistaciae*, insect habitats, integrated pest management, cone traps

## Abstract

**Simple Summary:**

The pistachio psyllid (*Agonoscena pistaciae*) is a harmful pest that causes significant damage to pistachio orchards. A study conducted in southern Türkiye between 2020 and 2022 revealed that the pest overwinters primarily in pine and cypress cones, as well as oak leaves near orchards. Cone-based traps were designed to take advantage of this behaviour and were found to capture approximately ten times more psyllids than natural cones. These findings provide the first evidence of overwintering in cones and suggest that cone-based traps could be an effective and environmentally friendly alternative to chemical pest control in pistachio orchards.

**Abstract:**

The pistachio psyllid (*Agonoscena pistaciae*) is a significant pest in pistachio (*Pistacia vera*) orchards, leading to serious economic losses. Understanding its overwintering behaviour is essential for developing effective pest control strategies. This study aimed to identify the overwintering habitats of *A. pistaciae* and to explore an alternative nature-based trapping method to managing its population. Field surveys were conducted over two years (2020–2022) in five key pistachio-growing regions of southern Türkiye. Our findings suggest that the *A. pistaciae* primarily overwinters in the mature cones of Turkish pine (*Pinus brutia*) and Mediterranean cypress (*Cupressus sempervirens*), as well as on the semi-evergreen leaves of oak trees (*Quercus brantii*, and *Q. infectoria*). Based on these observations, we developed cone-based overwintering traps and deployed them in pistachio orchards. These traps captured ten times more psyllids than those that were naturally overwintering in cones, which highlights their potential as a pest management tool. This study provides the first evidence of *A. pistaciae* overwintering in conifer cones and suggests that cone-based traps could serve as a practical and eco-friendly alternative to chemical control methods. Implementing this strategy in pistachio orchards may help reduce psyllid populations while preserving the ecological balance.

## 1. Introduction

Plants are constantly influenced by biotic and abiotic factors [1]. In recent years, the use of chemical pesticides to increase agricultural yields has allowed pests to diversify and develop resistance [2]. Pests of pistachio, particularly among horticultural crops, have increased and diversified. Pistachio (*Pistacia vera* L.) is an important crop, particularly in arid and semi-arid regions, due to its high commercial value and drought resistance [3]. Türkiye ranks among the top global producers of pistachios, having expanded its cultivation areas significantly over the past decade. Between 2010 and 2019, the total area of pistachio orchards increased from around 220,000 to over 366,200 hectares, representing a 63% increase in production capacity [4]. The southeastern Anatolia region accounts for over 90 per cent of Türkiye’s pistachio orchards, making it a key area for research on pest management strategies [5].

Of the many threats to pistachio production, the common pistachio psyllid (*Agonoscena pistaciae*) [6] is one of the most destructive pests. This insect is widely distributed across Türkiye, Greece, Syria, Iran, and other Mediterranean and Middle Eastern regions [7]. The nymphs and adults of *A. pistaciae* cause significant damage by piercing leaf tissues and extracting sap, which leads to physiological stress and reduced plant vigour. The honeydew secreted during and after feeding promotes the growth of sooty mould, which interferes with photosynthesis and reduces the overall health of the trees [8,9]. During severe infestations, premature leaf drop can occur, disrupting the formation of fruit buds and consequently leading to considerable economic losses in subsequent seasons [10]. Traditional pest-based control measures are applied to orchards during infestations However, genetic studies have revealed significant diversity within psyllid populations, which contributes to their adaptive capacity and resistance to control methods [11].

For decades, the primary method of controlling *A. pistaciae* has been the application of chemical insecticides [9,12]. Some of the most commonly used pesticides against *A. pistaciae* in Türkiye are Spirotetramat, Acetemiprid, and Lambda-cytotrin. The first two are systemic, while the latter is contact-active. However, as is often the case with other pests, excessive application of pesticides has led to the development of resistance, significantly reducing their effectiveness [13,14]. Furthermore, continuous pesticide applications disrupt beneficial insect populations, including predatory coccinellids and parasitoid wasps, which play a vital role in reducing psyllid outbreaks [15,16]. In some pistachio orchards in Türkiye and Iran, despite multiple pesticide applications per growing season sometimes exceeding six treatments, psyllids remain highly populous [17]. These challenges highlight the urgent need for alternative, sustainable pest management strategies [18]. The implementation of integrated pest management (IPM) practices has been proposed as a viable solution, focusing on reducing chemical reliance while promoting ecological balance [19].

One promising IPM approach is to study and understand *A. pistaciae*’s overwinteringbehaviour. A. pistaciaes’s overwintering behaviour is a critical survival strategy that influences insect population dynamics and subsequent infestation levels. Previous studies have suggested that psyllid species often migrate to nearby coniferous or semi-evergreen trees in search of shelter over winter [20]. While some studies indicate that *A. pistaciae* overwinters within pistachio orchards, particularly in tree bark and leaf litter [21], its preference for overwintering sites in surrounding natural vegetation remains unclear.

This study has two main objectives: (i) to identify the primary overwintering habitats of *Agonoscena pistaciae* a major pest that significantly reduces pistachio productivity in Türkiyeand (ii) to develop and evaluate an innovative biotechnical control method, specifically a cone-based trapping system, for reducing overwintering psyllid populations prior to the growing season.

Previous findings indicate that there is insufficient information available on the overwintering of *Agonoscena pistaciae*. Based on this ecological knowledge, cone-based overwintering traps have been developed and tested, showing great promise in reducing psyllid populations. This approach offers a more environmentally friendly alternative to chemical control and supports the sustainable management of *A. pistaciae* in pistachio orchards.

## 2. Materials and Methods

### 2.1. Preliminary Observations and Identification of Overwintering Areas of Agonoscena pistaciae

To determine the potential wintering areas of *A. pistaciae*, various samples such as tree bark, fruit, semi-evergreen oak leaves and pine cones were collected from different trees, shrubs and herbaceous plants growing near pistachio orchards in the study area in Gaziantep Province on 27 November 2020. The samples were bagged separately and marked from different locations. The spatial coordinates of each sample group were recorded. The samples were then transported to the Forest Entomology Laboratory at Kahramanmaraş Sütçü Imam University’s, Faculty of Forestry, in transparent polyethylene bags. The samples were examined under a stereo microscope for the presence of *A. pistaciae*. After preliminary observations revealing the significant presence of *A. pistaciae* adults in *Pinus brutia*, *Cupressus sempervirens* and *C. arizonica* cones and in semi-evergreen oak leaves for the first time within this study several field trips were organised to Gaziantep, Adıyaman, Şanlıurfa, Kilis, and Kahramanmaraş provinces between 30 November 2020 and 7 April 2021 (Figure 1) in an effort to ascertain whether this phenomenon is widespread. These provinces account for most of the country’s pistachio production. While Kahramanmaraş and Kilis are located in the Eastern Mediterranean region, Gaziantep, Şanlıurfa and Adıyaman are located in the Southeastern Anatolia region of Türkiye, where a hot dry climate prevails. During these trips, semi-evergreen oak leaves and cones were collected from oak and red pine forests mixed with pistachio trees, as well as from existing red pine and cypress trees surrounding schoolyards, cemeteries, mosques, official buildings, gas stations and other private fruit gardens. Samples were generally collected from the lower branches of the trees. The coordinates of the collection sites for all samples are given in Appendix A.

The study was conducted in the Turkish provinces of Gaziantep, Şanlıurfa, Kilis, Adıyaman, and Kahramanmaraş, which are the country’s main pistachio production areas to determine the wintering areas of *A. pistaciae*. These regions are extremely poor in terms of forest cover and play an important role in afforestation studies due to severe summer droughts and relatively low summer rainfall. Despite attempts to plant red pine (*Pinus brutia*) in designated areas in the early 2000s, forest cover remains limited to around 10% of the region as a whole [22]. Cypress trees (*Cupressus sempervirens* var. pyramidalis and *C. arizonica*) are widely planted along roadsides, near mosques, cemeteries, fields, public parks, and gardens throughout Türkiye and serve various functions such as providing agricultural support, protecting water sources conserving soil, and facilitating recreational activities. In the study area, pistachios and olives are widely cultivated in Southern Anatolia due to their tolerance of the region’s harsh environmental conditions.

### 2.2. Trap Preparation and Field Setup

All traps were installed on the branches of pistachio trees at 2 m on 2 August 2021 and retrieved on 22 December 2021. The insects were extracted in the laboratory and identified under a stereo microscope by Dr Daniel Burckhardt (Natural History Museum Basel, Switzerland) based on morphological characteristics [6,23]. All specimens were confirmed as *Agonoscena pistaciae*.

Once it was determined that *A. pistaciae* adults overwinter in pine and cypress cones, traps were prepared using these cones. To test whether these traps would attract *A. pistaciae* pests, they were set up in two different gardens in Kahramanmaraş province. The wintering *A. pistaciae* pests were identified by Dr. Daniel Burckhardt (Head of Entomology at the Natural History Museum Basel, Switzerland).

Observations indicated that *A. pistaciae* overwintered on the cones of Turkish pine and Mediterranean cypress. As an alternative to chemical interventions, a nature-based pest management approach was adopted using simple, low-cost traps to explore alternative methods for integrated pest management. Eight Turkish pine cone traps and twenty-four cypress cone traps, each containing 30 cones, were deployed on pistachio tree branches at two meters above the ground. Traps were installed on the private property site on 13 September 2021 and on the KSU site on 2 September 2021. A total of 480 Turkish pine cones and 1440 cypress cones were used as wintering traps. Cypress cones were placed by cutting the side areas of plastic bottles. Plastic bottles were modified to create cone traps, with rectangular openings allowing for easy insect entry (Figure 2). For statistical analysis, the traps and study sites were divided into four groups: private property—Turkish pine trap, KSU-Turkish pine trap, private property—cypress cone trap, and KSU—cypres—s cone trap (Table 1).

### 2.3. Collection of Insects from Cone Traps

The pine cone traps which had been designed for this purpose were hung on the lower branches of pistachio trees two metres above the ground in privately owned gardens (8 Turkish pine cone traps and 24 cypress cone traps) and in the KSU application gardens (8 Turkish pine cone traps and 24 cypress cone traps) on 2 September 2021. The traps were removed on 22 December 2021.

Thirty cone samples were placed in individual plastic bags and transported to the KSU Entomology Laboratory. The cone traps were then removed using diagonal pliers and gently shaken to dislodge the psyllids. The captured *A. pistaciaes* were collected in plastic bottles placed over white cardboard to facilitate counting. The cypress cones were stored in transparent polyethylene bags, and all insects inside were later examined under a stereo microscope. The total number of *A. pistaciaes* and other insect species was recorded.

### 2.4. Statistical Analysis

The traps were divided into four groups based on study site and tree species: Means, variances and standard deviations were calculated and are shown in Table 2. The count data for *A. pistaciae* and *Coccinellidae* showed overdispersion (variance ≫ mean), so Negative Binomial GLMs (glm.nb, MASS package in R) were used with property status and trap type (pine/cypress) as fixed effects and their interaction included.

## 3. Results

### 3.1. Overwintering A. pistaciae in Semi-Evergreen Oak Leaves

Observations revealed that *A. pistaciae* overwintered on the leaves of semi-evergreen oak trees, particularly *Quercus brantii* and, to a lesser extent, *Quercus infectoria*, in or near pistachio orchards in the study area. Although these oak leaves dried and turned brown, they remained attached to the trees until early spring. A substantial number of *A. pistciaes*’s natural enemies were also found on oak leaves including *Piocoris luridus*, Thomisidae spiders, Anthocoris sp., Chrysoperla sp., *Coccinellidae* adults, and parasitoid wasps. A significant number of psyllids were also found on fallen pistachio leaves.

### 3.2. Overwintering A. pistaciae in Cones

Specimens of *Agonoscena pistaciae* were collected from mature conifer cones. These cones primarily develop in Turkish pines after they reach 35–40 years of age and were found in abundance in older forested areas that were planted in the 1980s. The collection sites, coordinates, and number of *A. psitaciae* observed in twenty-five conifer cones are provided in the Appendix A. Analysis revealed that *A. pistaciae* populations were highest in Turkish pines aged 40 years or older, as these trees had retained their seeds from the previous season.

During field studies, Turkish pine cones harbouring the highest concentrations of *A. pistaciae* were ground and subjected to extraction analysis. The analysis revealed that the extractive substance content of the Turkish pine cones with the highest concentrations of *A. pistaciae* was 4.82%. The cones also showed signs of ageing, having lost their natural lustre due to prolonged exposure to environmental conditions. Various insect species were found in these cones, in addition to psyllid including spiders, brown lacewings, Psocoptera nymphs, mites, Coccinellidae, and Anthocoridae. Some *A. pistaciaes* had also been preyed upon by spiders.

Figure 3a shows the total number of *A. pistaciaes* and other insect species captured in the traps. This grouped bar chart shows the total number of *A. pistaciaes* and other species observed in the different traps. Figure 3a,b show the total count of each species in each trap. The differences in species abundance are shown. Additionally, Figure 3b shows the average distribution of beneficial insects across all traps. The pie chart illustrates the relative proportions of *Coccinellidae*, *Thomisidae, Piocoris* sp., and other beneficial species found in the collected samples.

Non *A. pistaciae* were detected in fallen conifer cones, although these cones were home to large populations of Collembola, Protura, Psocoptera, and mites. These cones, which had recently shed or were actively shedding seeds, contained 15 percent. No *A. pistaciaes* were observed in thirty different cone samples collected from overwintering sites. However, these cones contained high numbers of *Coccinellidae* larvae and adults, *Anthocoridae nymphs* and adults, *Deraeocoris* sp., *Piocoris luridus*, *Neuroptera* larvae and adults, *Bruchidae*, and various spider species (Table 2).

### 3.3. Analysis of Insects Collected from Cones

*A. pistaciae* specimens in each trap were examined and quantified under a stereo microscope. The results of the coniferous wood analysis showed that there were significant differences between the samples collected according to different cones (Figure 4) and location (Figure 5b).

The highest number of *Coccinellidae* individuals was recorded in trap 2, with an average of 35.50, followed by trap 4 (26.00), trap 1 (14.25), and trap 3 (13.04). Similarly, Thomisidae spiders were most abundant in trap 2, with an average of 38.50 individuals, followed by trap 1 (34.13), trap 4 (33.17), and trap 3 (19.46). Thomisidae spiders, known predators of psyllid adults, were observed preying on *A. pistaciaes*.

Anthocoridae numbers were highest in trap 4 (2.08), followed by trap 2 (1.88), trap 1 (1.88), and trap 3 (0.75). These insects are key predators of psyllid nymphs and are crucial for biological pest control. Piocoris individuals were most abundant in trap 2 (17.38), followed by trap 4 (7.88), trap 1 (0.38), and trap 3 (0.25). Similarly, Deraeocoris was only observed in traps 2 (1.50) and 4 (0.88), with no specimens found in traps 1 or 3 (Table 2).

The comparison of psyllid numbers across different trap types revealed significant differences. Figure 4 illustrates the variation in psyllid abundance between Turkish pine and cypress traps, highlighting a clear trend in psyllid preference for Turkish pine cones over cypress cones.

### 3.4. Statistical Analysis

Count data for *A. pistaciae* and other insects exhibited overdispersion (variance ≫ mean), violating the assumptions of Poisson and standard ANOVA models. Therefore, Negative Binomial generalized linear models (NB-GLMs) were fitted using the glm.nb [24,25,26,27,28,29] function of the MASS package in R [24]. The model included property (private property vs. KSU treatment garden) and trap type (pine vs. cypress) as fixed factors and their interaction.

The NB-GLMs revealed a highly significant interaction between altitude and trap type for *A. pistaciae* captures (β = 1.114, SE = 0.277, z = 4.025, *p* < 0.001), indicating that trap efficacy depended on location. In contrast, the abundance of other insects was influenced by main effects only: significantly lower at low altitude (β = –0.690, *p* < 0.001) and higher in pine cone traps (β = 0.311, *p* = 0.004), with no significant interaction (*p* = 0.200).

Pairwise comparisons (Tukey-adjusted, emmeans package; [30] showed that trap 1 (private orchards with pine) captured significantly more *A. pistaciae* than all other trap types (*p* < 0.001 in all cases). No other pairwise differences among **A. pistaciae** counts were significant (**p** > 0.145).

Full model outputs and pairwise *p*-values are presented in Table 3 and Table 4.

Further analysis of the distribution of beneficial insect species across different trap types is presented in Figure 6, which provides an overview of species abundance in each trap category. The heatmap highlights the dominance of *Coccinellidae* and *Thomisidae* spiders among beneficial insects, reinforcing their role in psyllid predation.

## 4. Discussion

This study has two main objectives: (i) to identify the overwintering habitats of *Agonoscena pistaciae*, a major pest of pistachios, and (ii) to develop innovative alternative wintering traps based on these findings. Previous research has demonstrated that *A. pistaciae* overwinters in various microhabitats, including tree bark, branches, and fallen leaves. However most studies have not specified particular plant species [31,32]. Our findings confirm that *A. pistaciae* prefers semi-evergreen oak leaves (*Quercus brantii*, *Q. infectoria*) and conifer cones (*Pinus brutia*, *Cupressus sempervirens*) as overwintering sites. This is the first report to confirm that psyllids actively use these materials for this purpose.

In Iran, research into the overwintering ecology of *A. pistaciae* has shown that this species of psyllid prefers old pistachio leaves and weed residues to bare soil for overwintering [33]. Similarly, studies in Türkiye have shown that predatory arthropods that overwinter emerge from artificial sites in pistachio orchards and adjacent habitats [34,35]. These results highlight the importance of alternative overwintering sites in regulating pest populations and suggest that *A. pistaciaes* can adapt to different winter shelters when natural refuges are scarce.

### 4.1. Impact of Agricultural Practices on Overwintering Psyllids

Throughout the research period (NovemberApril), pistachio orchards in the study region were closely monitored. Farmers commonly plough orchard soils to a depth of 30 cm before winter to eliminate *A. pistaciaes* overwintering within fallen leaves. Lenth [30] reported that similar orchard management strategies significantly reduce *A. pistaciae* populations by destroying overwintering refuges. However, these practices also force psyllids to seek alternative shelters, such as nearby forests and non-cultivated areas. Our study supports this observation, as we found that *A. pistaciae* populations were higher in forested areas adjacent to pistachio orchards, particularly in older pine stands with abundant cones.

### 4.2. Development and Efficiency of Alternative Wintering Traps

Research on alternative overwintering traps has demonstrated that the material, shape and colour of the trap can influence the efficiency with which insects are captured [36]. In northern China, shelter traps made from black corflute and wooden beehive structures were found to capture the highest number of overwintering insects [37]. Similarly, in Türkiye, pruning residues have been used as an effective alternative trap for managing the pistachio bark beetle, *Hylesinus vestitus*, reducing pest populations by 95–98% within two years [34].

Our study found that placing overwintering traps on pistachio trees successfully attracted *A. pistaciae*, particularly when the traps were positioned near the insects’ natural overwintering sites. This suggests that psyllid populations could be significantly reduced by deploying and removing these traps before the growing season. Using artificial overwintering traps could provide a sustainable, non-chemical approach to controlling *A. pistaciae*, complementing biological control methods.

### 4.3. Variability in Psyllid Overwintering Sites

Populations of *A. pistaciae* collected from Turkish pine cones varied significantly across regions. The highest densites of *A. pistaciae* were observed in forests adjacent to pistachio orchards in Gaziantep, whereas numbers were considerably lower in the other five trial areas. This discrepancy may be explained by differences in forest age; older forests (at least 20 years old) produce more cones, providing a greater number of suitable overwintering sites. Similar findings have been reported for other psyllid species, where host plant quality and availability strongly influence overwintering behaviour [33].

Conversely, the number of *A. pistaciae* in younger pine forests (aged ~20 years) was much lower, likely due to limited cone production and higher stand density. Forest management practices such as selective thinning could enhance cone production and increase the availability of overwintering habitats in the future [34].

Additionally, the distribution of *A. pistaciae* varied significantly between different sites. For instance, relatively few *A. pistaciae* were found in Turkish pine cones near Atatürk Dam, whereas higher numbers were recorded in cypress cones from the same region. This pattern may be influenced by differences in local pest communities and environmental conditions.

### 4.4. Implications for Integrated Pest Management (IPM)

Interviews with farmers revealed that control of the psyllid pest has largely relied on intensive pesticide applications, with some orchards receiving six to seven treatments per season. Similar trends have been reported in Iran, where excessive pesticide use has led to resistance in *A. pistaciae* populations and disrupted biological control mechanisms [7].

Our findings highlight the necessity of alternative pest management strategies that combine various methods, such as habitat manipulation, biological control, and targeted trapping systems. Conserving natural enemies such as coccinellid beetles and anthocorid bugs could enhance the effectiveness of non-chemical pest control methods [16].

## 5. Conclusions

The aim of this study was to develop effective management strategies for *Agonoscena pistaciae* populations by identifying their overwintering habitats and evaluating the effectiveness of specialised traps for controlling them. The findings confirm that *A. pistaciae* primarily overwinters in conifer cones (*Pinus brutia* and *Cupressus sempervirens*) and semi-evergreen oak leaves (*Quercus brantii*, *Q. infectoria*). These results emphasise the critical role that adjacent forested areas play in natural pest control, as these areas provide not only overwintering sites for *A. pistaciaes* but also habitats for their natural predators.

Statistical analyses revealed significant differences in the efficiency of the four types of overwintering traps, demonstrating that trap design and placement have a significant impact on their effectiveness. Although some traps contained fewer *A. pistaciaes*, the presence of beneficial predatory species such as coccinellids, anthocorids, and lacewings suggests that a natural predatorprey balance exists. Maintaining this balance is essential for managing *A. pistaciae* populations and ensuring the long-term sustainability of pistachio production.

One of the key findings of this study is that *A. pistaciaes* exhibit a strong preference for conifer cones, making them a viable option for targeted pest control. By placing these cones strategically in orchards in autumn, *A. pistaciaes* can be drawn into designated trapping areas, reducing their numbers before they can cause significant damage during the growing season. This method is consistent with previous research emphasising the importance of habitat modification in integrated pest management (IPM) strategies [7,34].

The results suggest that using overwintering traps is a more sustainable and environmentally friendly alternative to chemical pesticides. A comprehensive pest management strategy integrating overwintering traps biological control and selective pesticide use could significantly reduce *A. pistaciae* populations while preserving beneficial insect communities. Furthermore, the presence of predatory arthropods in forested areas near pistachio orchards supports the idea that these ecosystems should be preserved as part of a broader pest control strategy.

To refine and expand upon these findings, future studies should optimise the design of overwintering traps to improve *A. pistaciae* capture efficiency while minimising unintended effects on non-target species expand research to different geographic regions to assess the effectiveness of conifer cone traps under varying climatic conditions investigate the impact of forest management practices (such as selective thinning and controlled burning) on *A. pistaciae* overwintering behaviour and population dynamics evaluate the economic feasibility of implementing conifer cone traps on a large scale in commercial pistachio orchardsand enhance conservation-based biological control efforts by promoting the presence of natural enemies of *A. pistaciae* through habitat preservation and reduced pesticide usage. This studyhighlights the potential of conifer cones as a valuable tool for managing *A. pistaciae* populations within an integrated pest management framework. Leveraging the *A. pistaciae*’s natural overwintering tendencies and strengthening the role of predatory arthropods could lead toa more sustainable and ecologically balanced pest control strategy. Implementing these findings in pest control programmes could reduce the need for chemical pesticides, mitigate economic losses and promote ecological stability in pistachio orchards. The strategic use of conifer cones in conjunction with biological control methods, represents a practical and environmentally responsible approach to controlling *A. pistaciae* populations in Türkiye and beyond.

This study confirms for the first time that *A. pistaciae* overwinters in conifer cones and semi-evergreen oak leaves, thereby expanding our understanding of its overwintering behaviour. Leveraging this knowledge, we developed and tested a novel trap system for overwintering, demonstrating its potential for suppressing *A. pistaciae* populations.

Future research should focus on optimising trap designs to improve efficiency and cost-effectiveness. Additionally, long-term studies assessing the impact of forest management practices, such as selective thinning and reforestation, on *A. pistaciae* population dynamics would enhance our understanding of pest ecology further. Expanding this study to other pistachio-growing regions would help to validate the applicability of these findings under different climatic and ecological conditions.

By integrating habitat modification, biological control, and targeted trapping systems, a more sustainable approach to *A. pistaciae* management can be developed, reducing reliance on chemical pesticides while maintaining pistachio orchard productivity.

## Figures and Tables

**Figure 1 insects-16-01150-f001:**
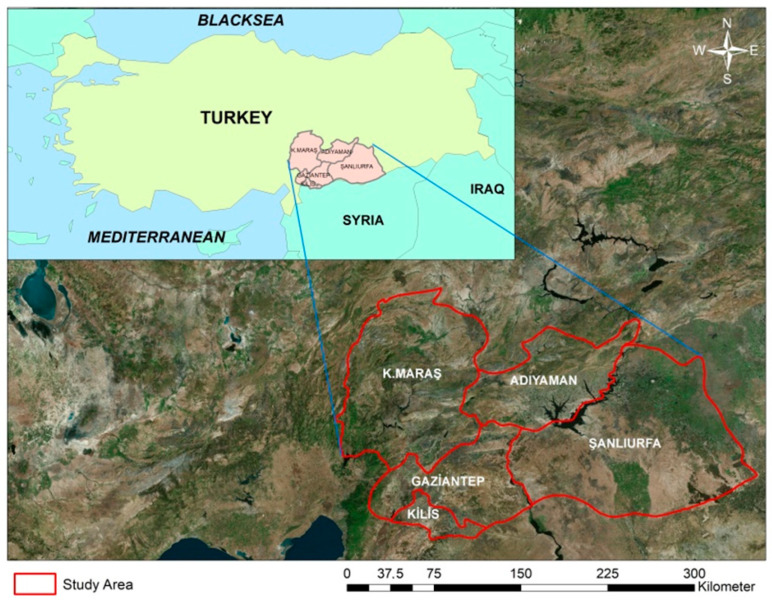
Location of the study areas in the Eastern Mediterranean and Southeastern Anatolia Regions, where more than 90% of pistachio production in Türkiye is produced.

**Figure 2 insects-16-01150-f002:**
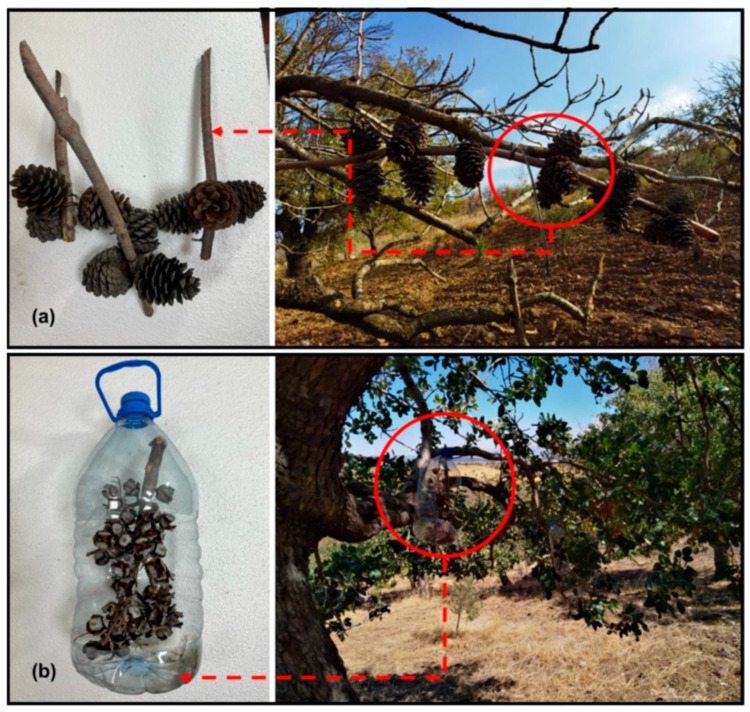
The designed overwintering traps and their installation on pistachio trees (**a**) Turkish pine and (**b**) Cypress cone-based traps.

**Figure 3 insects-16-01150-f003:**
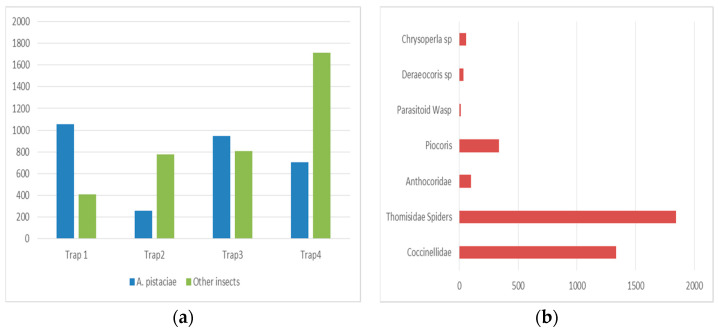
(**a**) A grouped bar chart showing the total number of observed insects in traps. *A. pistaciae* (blue) and other species (red) are displayed clearly for comparison. (**b**) A single, aggregated pie chart representing the average distribution of beneficial insects across all traps. The categories *Coccinellidae*, *Thomisidae*, *Piocoris* sp., and others (<5%) are shown with percentages.

**Figure 4 insects-16-01150-f004:**
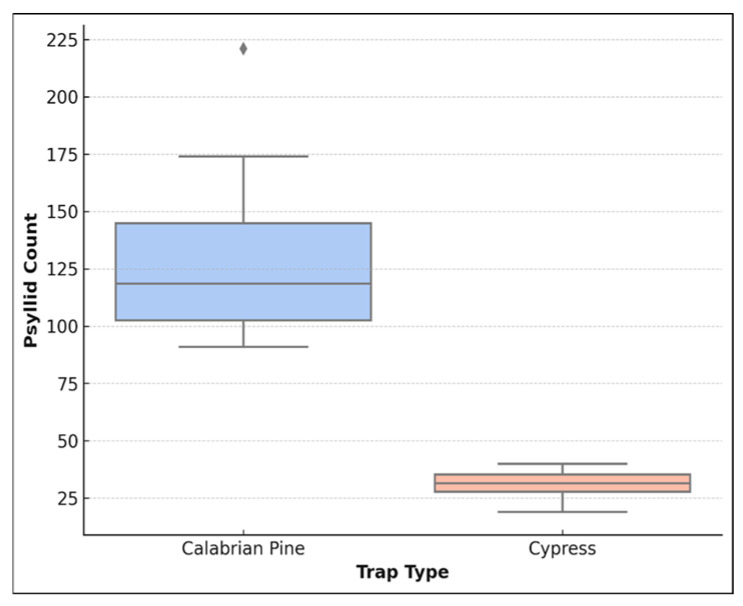
*A. pistaciae* of psyllid count distribution in different trap types.

**Figure 5 insects-16-01150-f005:**
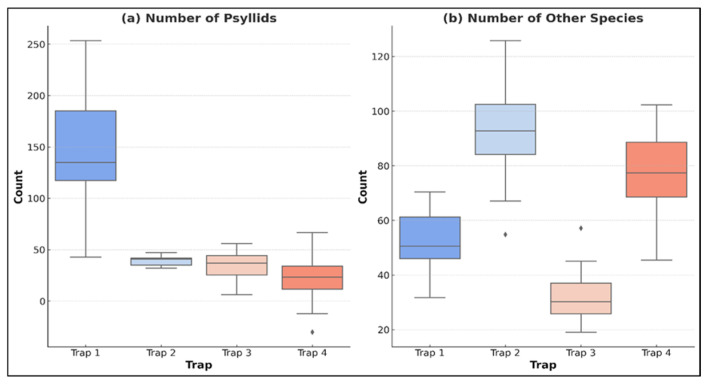
Box-whisker plot for *A. pistaciae* (**a**) and other insect species (**b**).

**Figure 6 insects-16-01150-f006:**
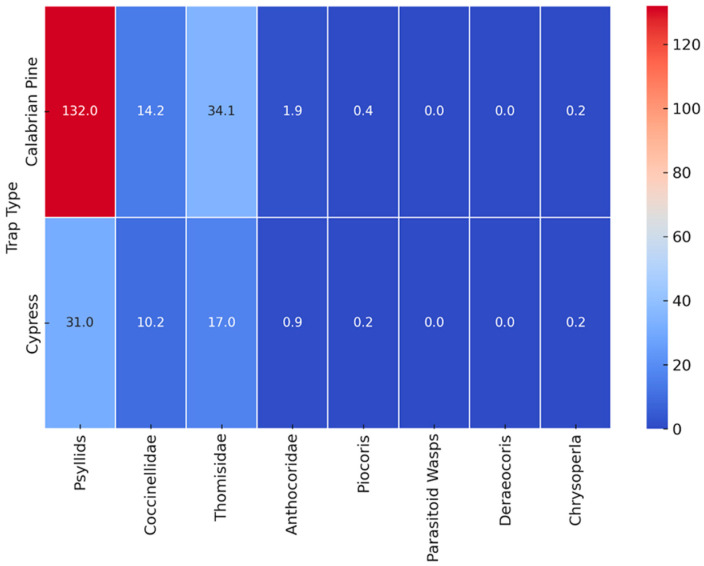
Average count of *A. pistaciae* and beneficial species by trap type.

**Table 1 insects-16-01150-t001:** The properties of traps.

Trap No.	Trap Type	Location	Altitude (m)
1	Turkish pine cone	Private property	652
2	Cypress cone	Private property	652
3	Turkish pine cone	KSÜ Orchard	966
4	Cypress cone	KSÜ Orchard	966

**Table 2 insects-16-01150-t002:** Descriptive statistics of Psyllid and other insect counts by trap type (1–4).

Species	Trap	N	Mean	Std. Deviation	Minimum	Maximum
Psyllid	1	8	132.00	44.81	91	221
	2	8	32.13	6.66	22	40
	3	24	39.50	18.94	17	112
	4	24	29.30	23.8	11	112
Others	1	8	14.25	2.49	11	19
	2	8	35.50	8.77	23	50
	3	24	13.04	4.66	6	22
	4	24	26.00	8.10	13	42

**Table 3 insects-16-01150-t003:** Parameter estimates from Negative Binomial GLMs for *Agonoscena pistaciae* and *Coccinellidae* captures.

Species	Term	Estimate (SE)	z	*p*-Value
*A. pistaciae*	(Intercept)—KSU, Cypress	3.377 (0.100)	33.632	<0.001
	Private orchard Cypress	0.299 (0.141)	2.125	0.034
	Trap Pine KSU	0.092 (0.200)	0.462	0.644
	Private orchard × Trap Pine	1.114 (0.277)	4.025	<0.001
Other insects	(Intercept)—KSU Cypress	3.258 (0.057)	57.215	<0.001
	Private orchard Cypress	–0.690 (0.090)	–7.677	<0.001
	KSU orchard TrapPine	0.311 (0.108)	2.881	0.004
	Private orchard × TrapPine	–0.223 (0.174)	–1.282	0.200

Reference group: KSU cypress. Private orchard = 966 m; high = 652 m.

**Table 4 insects-16-01150-t004:** Tukey-adjusted *p*-values for pairwise comparisons among trap groups.

Comparison	*A. pistaciae* (*p*)	*Coccinellidae* (*p*)
KSU Cypress—Private Cypress	0.145	<0.001
KSU Cypress—KSU Pine	0.967	0.021
KSU Cypress—Private Pine	<0.001	<0.001
Private Cypress—KSU Pine	0.727	<0.001
Private Cypress—Private Pine	<0.001	0.915
KSU Pine—Private Pine	<0.001	<0.001

## Data Availability

The original contributions presented in this study are included in the article. Further inquiries can be directed to the corresponding authors.

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
