# Peer review of "Overwintering Ecology and Novel Trapping Strategies for Sustainable Management of the Common Pistachio Psyllid (*Agonoscena pistaciae*) in Pistachio Orchards"

_insects, 2025, doi:10.3390/insects16111150_

Round 1

Reviewer 1 Report

Comments and Suggestions for Authors

The pistachio psyllid, Agonoscena pistaciae is an important pest of pistachio in Turky. As the author discusses, excessive reliance on chemical pesticides to control this pest may be a threat to sustainable pistachio production in the country. The author examined the overwintering ecology of A. pistaciae, and based on the derived findings, he developed a new, efficient, and eco-friendly trap to monitor the occurrence of this pest. This kind of work will encourage us to promote less harmful pest control as represented by IPM. Overall, I had positive impression on this work, however, I need to suggest some substantial revision to the manuscript, particularly about the statistical analyses.

The author divided his trap data into four groups by 2 trap types and 2 locations. However, he performed one-way ANOVA to see effects of traps. Why didn’t he perform two-way ANOVA. It is the best analysis to see effects of trap type, location, and their interaction. Furthermore, from Lines 241-243, the author declared that one of the four groups (trap 4) was excluded from statistical analysis. As long as I read this part, I could not understand the validity of this process: differences in mean values and SD must be corrected by log transformation, which has been already done (Line 174); what was non-homogeneous, species, number of trapped insects? I strongly recommend including the data of trap 4 into the statistical analyses and discussion based on the results.

Please clarify that ‘psyllids’ mentioned in this manuscript means A. pistaciae only, not a mixture of Psyllids insect species. It is pretty important because this paper focuses on A. pistaciae. According to Line 141-142, Dr. Daniel Burckhardt identified A. pistaciae trapped during winter. How did he identify the insects, morphology or DNA? Please cite a reference that he used to identify them. Also, did he also identify all pshyllids trapped in this work (including summer-autumn trapping described in section 2.4)?

<Specific comments>

Line 63: please provide some chemicals that are usually applied to control this pest.

Line 99: double space.

Line 108 and Appendix 1: date was shown in ‘dd.nn.yyyy’ here, but ‘month dd, yyyy’ for other parts of the manuscript. Please unify.

Line 108: remove ‘.)’.

Lines 121-124: remove this sentence. It was already shown in Lines 110-112.

Line 154: please insert ‘(Table 1)’ at the end of this sentence.

Lines 159-161: Please clarify number of traps used in the summer-autumn investigation.

Line 185: remove this line.

Lines 193-194: I couldn’t understand from M&M how the author obtained this data.

Line 201: ‘clearly illustrating’ is not objective, need statistical confirmation.

Line 211 and 232: Table 2 did not show what the author described here.

Line 216 (Table 2): no foot notes to explain the meanings of asterisk and A-C.

Lines 218-220: These two sentences have already been shown in M & M.

Lines 221 and 233: please show statistical values to confirm ‘significant differences’  

Line 222: ‘Fig. 5a,b’.

Line 224: What is the unit of these values, individuals per day, or per trap?

Line 249: ‘p < 0.00’ is not appropriate. ‘p < 0.01’ or ‘p < 0.001’.

Line 250: please show actual P value when it was higher than 0.01 (or 0.001).

Lines 263, 266, 270 and elsewhere: show ‘A. pistaciae’ in italic.

Lines 332-390 (Conclusions): this is too long as conclusion. Please summarize in a few sentences or make a new section as a part of Discussion.

Line 340: The author compared only three types of traps statistically.

Author Response

Dear Reviewer 1,
Thank you for your valuable contributions. Your suggestions have improved the readability and value of the article. I have revised my paper step by step based on your comments. 

Comments 1: The author divided his trap data into four groups by 2 trap types and 2 locations. However, he performed one-way ANOVA to see effects of traps. Why didn’t he perform two-way ANOVA. It is the best analysis to see effects of trap type, location, and their interaction. Furthermore, from Lines 241-243, the author declared that one of the four groups (trap 4) was excluded from statistical analysis. As long as I read this part, I could not understand the validity of this process: differences in mean values and SD must be corrected by log transformation, which has been already done (Line 174); what was non-homogeneous, species, number of trapped insects? I strongly recommend including the data of trap 4 into the statistical analyses and discussion based on the results.

Response 1: Based on the results, the data from trap 4 were included in the statistical analyses and discussions.

Comments 2: Please clarify that ‘psyllids’ mentioned in this manuscript means A. pistaciae only, not a mixture of Psyllids insect species. It is pretty important because this paper focuses on A. pistaciae.

Response 2: The word psyllid was replaced by A. pistaciae.

Comments 3: According to Line 141-142, Dr. Daniel Burckhardt identified A. pistaciae trapped during winter. How did he identify the insects, morphology or DNA?

Response 3: He identified insects based on their morphology.

Comments 4: Please cite a reference that he used to identify them. Also, did he also identify all pshyllids trapped in this work (including summer-autumn trapping described in section 2.4)?

Response 4: No, he only described overwintering A. pistaciae.

Count data for A. pistaciae and other insects exhibited overdispersion (variance ≫ mean), violating the assumptions of Poisson and standard ANOVA models. Therefore, Negative Binomial generalized linear models (NB-GLMs) were fitted using the glm.nb() function from the MASS package in R (Venables & Ripley, 2002). The model included Property (Private Property vs. KSU treatment garden) and Trap type (Pine vs. Cypress) as fixed factors and their interaction.

The NB-GLM revealed a highly significant interaction between altitude and trap type for A. pistaciae captures (β = 1.114, SE = 0.277, z = 4.025, p < 0.001), indicating that trap efficacy depended on location. In contrast, other insects abundance was influenced by main effects only: significantly lower at low altitude (β = –0.690, p < 0.001) and higher in pine cone traps (β = 0.311, p = 0.004), with no significant interaction (p = 0.200).

Post-hoc pairwise comparisons (Tukey-adjusted, emmeans package; Lenth, 2023) showed that Trap 1 (Private orchards Pine) captured significantly more A. pistaciae than all other trap types (p < 0.001 in all cases). No other pairwise differences among *A. pistaciae* counts were significant (*p* > 0.145).

Full model outputs and pairwise p-values are presented in Tables 2 and 3.

Please clarify that ‘psyllids’ mentioned in this manuscript means A. pistaciae only, not a mixture of Psyllids insect species. It is pretty important because this paper focuses on A. pistaciae. According to Line 141-142, Dr. Daniel Burckhardt identified A. pistaciae trapped during winter. How did he identify the insects, morphology or DNA? Please cite a reference that he used to identify them. Also, did he also identify all pshyllids trapped in this work (including summer-autumn trapping described in section 2.4)? All of them revised.

  • Line 63: please provide some chemicals that are usually applied to control this pest.; The names of the 3 most commonly used pesticides have been added. Some of the most commonly used chemical pesticides against A. pistaciae in Türkiye are Spirotetramat, Acetemiprid, and Lambda-cytotrin. The first two are systemic, while the other is contact-action
  • Line 99: double space;
  • Line 108 and Appendix 1: date was shown in ‘dd.nn.yyyy’ here, but ‘month dd, yyyy’ for other parts of the manuscript. Please unify. Corrected.
  • Line 108: remove ‘.)’.
  • Lines 121-124: remove this sentence. It was already shown in Lines 110-112.
  • Line 154: please insert ‘(Table 1)’ at the end of this sentence. Added
  • Lines 159-161: Please clarify number of traps used in the summer-autumn investigation. The number of traps used in the summer-autumn survey has been announced.
  • Line 185: remove this line. Removed
  • Lines 193-194: I couldn’t understand from M&M how the author obtained this data. Announced
  • Line 201: ‘clearly illustrating’ is not objective, need statistical confirmation. The word clearly was removed from the sentence
  • Line 211 and 232: Table 2 did not show what the author described here.
  • Line 216 (Table 2): No foot notes to explain the meanings of asterisk and A-C. Letter groups (A, B, C) are missing. All p values ​​are clear. Separate columns for the two types.
  • 218-220: These two sentences have already been shown in M & M. Corrected The total number of psyllids and other insect species captured in the traps is presented in Figure 3a.”
  • 221 and 233: please show statistical values to confirm ‘significant differences’ → “Boxplots of psyllid counts by trap type are shown in Figure 4.” → “The distribution of beneficial insects across trap types is visualized as a heatmap in Figure 6.” → “Box-whisker plots for (a) A. pistaciae and (b) other insect species are shown in Figure 5.”
  • 5a,b’.Corrected “Fig. 5a,b” → “Figure 5”
  • What is the unit of these values, individuals per day, or per trap?
  • Line 249: ‘p < 0.00’ is not appropriate. ‘p < 0.01’ or ‘p < 0.001’.
  • Line 250: please show actual P value when it was higher than 0.01 (or 0.001).
  • Lines 263, 266, 270 and elsewhere: show ‘A. pistaciae’ in italic.
  • Lines 332-390 (Conclusions): this is too long as conclusion. Please summarize in a few sentences or make a new section as a part of Discussion.
  • Line 340: The author compared only three types of traps statistically.

Reviewer 2 Report

Comments and Suggestions for Authors

This manuscript presents a highly valuable and applied study on the overwintering ecology of a major pistachio pest, Agonoscena pistaciae, and the development of a novel, eco-friendly trapping system based on this ecological insight. The research addresses a significant agricultural problem and aligns perfectly with the growing need for sustainable Integrated Pest Management (IPM) strategies. The discovery of conifer cones as a primary overwintering habitat is novel and important. The translation of this basic ecological finding into a practical management tool (cone-based traps) is a particular strength of the work.

The experimental design is robust, encompassing a two-year survey across multiple key pistachio-producing regions. The results are promising, demonstrating a significant capture efficacy of the designed traps. However, the manuscript requires major revisions before it can be considered for publication. The most critical issues lie in the presentation and clarity of the statistical analysis and methodological details, which currently hinder a full assessment of the results. Additionally, the manuscript requires thorough proofreading for language and flow.

Clarity in Methods (Section 2.3 & 2.4):

The trap design is mentioned (modified plastic bottles) and shown in Figure 2, but a more detailed description or a reference to a diagram would be helpful. How exactly were the cones placed inside?

Statistical Analysis Presentation (Section 2.5, 3.3, 3.4):

Table 2 is problematic. It lists only three traps (1, 2, 3), but the methods describe four groups (see Table 1). What happened to Trap 4 (Cypress cone at KSU)? The text in 3.4 mentions excluding Trap 4 from ANOVA due to non-homogeneous distribution, but this exclusion and the rationale need to be clearly stated in the results section before presenting the data. The data in Table 2 currently do not match the experimental design.

I noticed an error in the figure numbering: what is referred to as Figure 3 in the text appears to be composed of Figure 4a and 4b.

The results mention a boxplot (Figure 4) and a heatmap (Figure 6), which are excellent, but the statistical values (F-statistics, p-values) in section 3.4 need to be explicitly linked to the comparisons being made (e.g., between which trap types?). Re-structure the results to clearly explain the fate of all four traps. Clearly state why Trap 4 was excluded from the formal ANOVA and how it was handled. Ensure all tables and figures correspond to the described experimental design. Integrate statistical results (e.g., "Trap 1 captured significantly more psyllids than Trap 2 and 3 (F = 113.67, p < 0.001; Duncan's test)"). Double-check all values in tables for consistency.

Writing and Flow:

The manuscript has minor grammatical errors and awkward phrasing (e.g., "preliminary observations revelling significant," "wintering areas such as tree barks, fruits," "Feb, 02 – 21, 2021" in Appendix).

Some information is repeated, particularly the description of the study regions in Sections 2.1 and 2.2.

The discussion and Conclusions is strong but could be more focused. The section "4.3. Variability in Psyllid Overwintering Sites" contains interesting points about forest age and competition, but some speculations (e.g., about bast scale insects) feel slightly out of place without data from this study to support them.

Author Response

Dear Reviewer 1,
Thank you for your valuable contributions. Your suggestions have improved the readability and value of the article. I have revised my paper step by step based on your comments. 

Comments 1: Clarity in Methods (Section 2.3 & 2.4):

The trap design is mentioned (modified plastic bottles) and shown in Figure 2, but a more detailed description or a reference to a diagram would be helpful. How exactly were the cones placed inside?

Response 1: Cypress cones were placed by cutting the side areas of plastic bottles.

Comments 2: Statistical Analysis Presentation (Section 2.5, 3.3, 3.4):

Table 2 is problematic. It lists only three traps (1, 2, 3), but the methods describe four groups (see Table 1). What happened to Trap 4 (Cypress cone at KSU)? The text in 3.4 mentions excluding Trap 4 from ANOVA due to non-homogeneous distribution, but this exclusion and the rationale need to be clearly stated in the results section before presenting the data. The data in Table 2 currently do not match the experimental design.

Response 2: Table 2 was revised to correct the problems. Four trap groups were also identified. The data in Table 2 was aligned with the experimental design.

Count data for A. pistaciae and other insects exhibited overdispersion (variance ≫ mean), violating the assumptions of Poisson and standard ANOVA models. Therefore, Negative Binomial generalized linear models (NB-GLMs) were fitted using the glm.nb() function from the MASS package in R (Venables & Ripley, 2002). The model included Property (Private Property vs. KSU treatment garden) and Trap type (Pine vs. Cypress) as fixed fac-tors and their interaction.

The NB-GLM revealed a highly significant interaction between altitude and trap type for A. pistaciae captures (β = 1.114, SE = 0.277, z = 4.025, p < 0.001), indicating that trap efficacy depended on location. In contrast, other insects abundance was influenced by main effects only: significantly lower at low altitude (β = –0.690, p < 0.001) and higher in pine cone traps (β = 0.311, p = 0.004), with no significant interaction (p = 0.200).

Post-hoc pairwise comparisons (Tukey-adjusted, emmeans package; Lenth, 2023) showed that Trap 1 (Private orchards Pine) captured significantly more A. pistaciae than all other trap types (p < 0.001 in all cases). No other pairwise differences among *A. pistaciae* counts were significant (*p* > 0.145).

Full model outputs and pairwise p-values are presented in Tables 2 and 3.

Comments 3: I noticed an error in the figure numbering: what is referred to as Figure 3 in the text appears to be composed of Figure 4a and 4b.

Response 3: Figure 3 corrected

Comments 4: The results mention a boxplot (Figure 4) and a heatmap (Figure 6), which are excellent, but the statistical values (F-statistics, p-values) in section 3.4 need to be explicitly linked to the comparisons being made (e.g., between which trap types?). Re-structure the results to clearly explain the fate of all four traps. Clearly state why Trap 4 was excluded from the formal ANOVA and how it was handled. Ensure all tables and figures correspond to the described experimental design. Integrate statistical results (e.g., "Trap 1 captured significantly more psyllids than Trap 2 and 3 (F = 113.67, p < 0.001; Duncan's test)"). Double-check all values in tables for consistency.

Response 4: All problems have been resolved

Comments 5: Variability in Psyllid Overwintering Sites" contains interesting points about forest age and competition, but some speculations (e.g., about bast scale insects) feel slightly out of place without data from this study to support them.

Response 5: The term bast scale insects has been removed.

Round 2

Reviewer 1 Report

Comments and Suggestions for Authors

I appreciate that the author sincerely made revision based on the comments from reviewer(s). All my concern in the first review were cleared, and the manuscript seems to be ready for publicaiton. I recommend the author to check English in the main text (e.g., Lines 274-275) and the apprearance of Tables and Figures (e.g., Table 2).

I hope that this study will contribute to better management of Agonoscena pistaciae in Türkiye.

Author Response

Commend 1. I recommend the author to check English in the main text (e.g., Lines 274-275) and the apprearance of Tables and Figures (e.g., Table 2).

Responds 1. Thank you for your positive feedback on the article. Associate Professor Dr Emre Babur edited the revised English version. Lines. 274-275

Tables and figures checked and revised

Reviewer 2 Report

Comments and Suggestions for Authors

I have no more questions here.

Author Response

Thank you for your positive feedback on the article. The revised English version was edited by Associate Professor Dr Emre Babur.